# The β-Blocker Carvedilol Prevents Benzo(a)pyrene-Induced Lung Toxicity, Inflammation and Carcinogenesis

**DOI:** 10.3390/cancers15030583

**Published:** 2023-01-18

**Authors:** Ayaz Shahid, Mengbing Chen, Carol Lin, Bradley T. Andresen, Cyrus Parsa, Robert Orlando, Ying Huang

**Affiliations:** 1Department of Pharmaceutical Sciences, College of Pharmacy, Western University of Health Sciences, Pomona, CA 91766, USA; 2College of Osteopathic Medicine of the Pacific, Western University of Health Sciences, Pomona, CA 91766, USA; 3Department of Pathology, Beverly Hospital, Montebello, CA 90640, USA

**Keywords:** carvedilol, lung cancer, benzo(a)pyrene, β-blocker, tobacco smoking

## Abstract

**Simple Summary:**

Benzo(a)pyrene is a ubiquitously present environmental contaminant that induces lung cancer after being converted into active metabolites via aryl hydrocarbon receptor (AhR)-mediated metabolic activation. Previous reports showed that carvedilol is one of the most effective β-blockers with skin cancer preventive activity. The current study evaluated the effect and mechanism of carvedilol on benzo(a)pyrene-induced lung toxicity, inflammation and carcinogenesis. Carvedilol blocked the benzo(a)pyrene-induced malignant transformation of human bronchial epithelial cells and inhibited the benzo(a)pyrene-induced activation of the oncogenic signaling of ELK-1, NF-κB and AhR. This result suggests that the cross-talk of these signaling pathways plays an important role in mediating both inflammation and carcinogenesis in the lung. Carvedilol’s activity was further confirmed on mouse models of acute lung toxicity and inflammation, as well as chronic lung cancer development induced by benzo(a)pyrene. As an FDA-approved generic drug, carvedilol may be repurposed as a lung cancer preventive agent.

**Abstract:**

The current study evaluated the effects of the β-blocker carvedilol on benzo(a)pyrene (B(a)P) and its active metabolite benzo(a)pyrene diol epoxide (BPDE)-induced lung toxicity, inflammation and carcinogenesis and explored the potential mechanisms. Carvedilol blocked the BPDE-induced malignant transformation of human bronchial epithelial cells BEAS-2B. In BEAS-2B cells, B(a)P strongly activated ELK-1, a transcription factor regulating serum response element (SRE) signaling, which was attenuated by carvedilol. Carvedilol also inhibited the B(a)P-induced AhR/xenobiotic responsive element (XRE) and mRNA expression of *CYP1A1* and attenuated B(a)P-induced NF-κB activation. In a B(a)P-induced acute lung toxicity model in CD-1/IGS mice, pretreatment with carvedilol for 7 days before B(a)P exposure effectively inhibited the B(a)P-induced plasma levels of lactate dehydrogenase and malondialdehyde, inflammatory cell infiltration and histopathologic abnormalities in the lung, and upregulated the expression of GADD45α, caspase-3 and COX-2 in the lung. In a B(a)P-induced lung carcinogenesis model in A/J mice, carvedilol treatment for 20 weeks did not affect body weight but significantly attenuated tumor multiplicity and volume. These data reveal a previously unexplored role of carvedilol in preventing B(a)P-induced lung inflammation and carcinogenesis by inhibiting the cross-talk of the oncogenic transcription factors ELK-1, AhR and NF-κB.

## 1. Introduction

Lung cancer is one of the leading causes of death in both genders worldwide, and approximately 350 deaths per day occurred from lung cancer in the United States in 2022 [1]. Chronic exposure to chemical carcinogens in tobacco smoke and air pollution is known as the major factor for lung cancer initiation and development [2]. While smoking cessation decreased cigarette usage in the US, the lung cancer risk for smokers remains high. As late-stage lung cancers are usually resistant to pharmacotherapies, one of the most important strategies to manage lung cancer is chemoprevention, which uses pharmacological agents to block or reverse cancer initiation, promotion and progression in the lung [3]. As no effective chemopreventive agent currently exists, the development of novel chemopreventive agents for high-risk populations, such as smokers, is urgently needed.

Among tobacco smoke constituents, polycyclic aromatic hydrocarbons (PAHs) play a significant role in mediating lung cancer development. [4,5]. The most well-studied PAH, benzo(a)pyrene (B(a)P), mainly detected in cigarette smoke and automobile exhausts, causes lung toxicity and the initiation of cancer after metabolic activation to benzo(a)pyrene diol epoxide (BPDE), which is one of the electrophilic reactive metabolites [5]. B(a)P-induced acute toxicity and tumor formation in the mouse lung provide clinically relevant models widely used in preclinical cancer prevention research [5]. B(a)P acts as a genotoxic and non-genotoxic carcinogen by damaging DNA and activating oncogenic signaling such as the mitogen-activated protein kinases (MAPKs), aryl hydrocarbon receptor (AhR) and NF-κB [5,6,7]. 

β-blockers, widely prescribed for cardiovascular disorders, work by inhibiting the interaction of catecholamines (epinephrine and norepinephrine) with the β-adrenergic receptors. Currently, β-blockers are being examined as cancer therapeutic adjuvants because the use of β-blockers has been linked with the prolonged survival of lung cancer patients and has been proposed for preventing postoperative lung cancer recurrence [8,9,10,11,12,13]. Additionally, various findings from animal models or human subjects suggest that β-blockers may reduce the incidence of lung cancer, although the results were not always consistent [14,15,16]. Furthermore, it has been reported that B(a)P binds directly to the β2-adrenergic receptor and uses this pathway to induce an increase in intracellular calcium, which contributes to the AhR-mediated toxic effects of B(a)P [17]. However, the effects of β-blockers on B(a)P-mediated toxic and carcinogenic effects in the lung have not been investigated. 

The β-blocker carvedilol, approved in 1995 by the U.S. FDA, is a receptor subtype non-selective β-blocker with antioxidant and anti-inflammatory properties [18,19,20]. Carvedilol has shown cancer preventive properties against chemical carcinogen 7,12-dimethylbenz[a]anthracene (DMBA)-induced skin hyperplasia [21] and ultraviolet (UV)-induced squamous carcinoma development in mice [22]. Among the multiple β-blockers examined, carvedilol showed the highest potency in preventing the malignant transformation of epidermal cells [23]. The preventive activity of carvedilol has also been demonstrated in human breast epithelial cells’ malignant transformation induced by B(a)P [24]. Recently, it also showed that in a population-based cohort study, patients who received carvedilol treatment resulted in a significantly lower rate of lung cancer compared to the same number of matched controls without carvedilol use [25]. Therefore, we hypothesized that carvedilol may prevent B(a)P-induced lung carcinogenesis. 

In the present study, we investigated the chemopreventive efficacy of carvedilol using in vitro and in vivo models induced by B(a)P or its active metabolite BPDE. First, we investigated the carvedilol’s effects on the malignant transformation of human bronchial epithelial cells and explored the mechanisms. We next demonstrated that carvedilol had a protective activity against the B(a)P-induced short-term biomarkers of oxidative stress and inflammation in the lung. Eventually, we demonstrated the chemopreventive efficacy of carvedilol in a lung cancer mouse model induced by B(a)P. These data provide experimental evidence that the β-blocker carvedilol may be used as a preventive treatment for individuals with an increased risk of lung cancer. 

## 2. Materials and Methods

### 2.1. Compounds

Carvedilol was purchased from Tocris Bioscience (Minneapolis, MN, USA) for cell culture work; the powder was reconstituted in dimethylsulfoxide (DMSO) as a stock solution of 10 mM, stored at −20 °C and diluted into culture media. For animal work, carvedilol was purchased from Santa Cruz Biotechnology (Dallas, TX, USA); the powder was dissolved in DMSO as a stock solution of 111 and 18 mM, stored at −20 °C and diluted into mouse drinking water which was prepared weekly. Benzo(a)pyrene was purchased from Sigma-Aldrich (St. Louis, MO, USA). Benzo(a)pyrene diol epoxide (BPDE) was purchased from Santa Cruz Biotechnology. 

### 2.2. Cell Line and Cell Culture 

BEAS-2B, an immortalized human bronchial epithelial cell line, was purchased from ATCC (cat #: CRL-9609). For soft agar and MTT assays, the cells were cultured in an airway epithelial cell growth medium kit containing a supplement mix (PromoCell, Heidelberg, Germany; VWR cat # 10175-240 or 10175-246). For luciferase and RT-PCR assays, the BEAS-2B cells were cultured in K-SFM (Invitrogen, Grand Island, NY, USA) supplemented with 5 ng/mL recombinant epidermal growth factor (EGF) and 50 μg/mL bovine pituitary extracts. For all in vitro assays, the cells were incubated at 37 °C in 95% air and 5% CO_2_ atmosphere until they approached 80% confluence. A 0.05% trypsin/EDTA solution (GenClone 25-510F Trypsin-EDTA, 0.05% 1×) was used for cell subculture.

### 2.3. Anchorage-Independent Growth Assay in Soft Agar

The soft agar assay for determination of cellular transformation or anchorage-independent growth was conducted according to a protocol described previously with modification [26]. In brief, the BEAS-2B cells at a density of 500 cells per cm^2^ were seeded in 75 cm^2^ dishes containing complete media. Then, 24 h after cell plating, the cells were treated with a single dose of BPDE (0.2 μM) for 1 h and washed with DPBS, which was replaced with complete media, and the cells were further cultured for 5 to 7 days when the cells were approximately 80% confluence. The cells were then plated for soft agar assay. In total, 2000 cells per well were mixed with 0.33% noble agar (Sigma) in complete media in a 96-well tissue culture plate, layered over a solidified bottom layer containing 0.5% agar in complete media. To evaluate the effects of carvedilol, the cells were pre-treated with various concentrations of the drug for 2 h before exposure to BPDE. After BPDE exposure, fresh media containing carvedilol were added to the culture. Carvedilol was also added into the top layer of the agar mixed with the cells. For 7–10 days, plates were incubated at 37 °C with 5% CO_2_/95% air. Colonies were counted under a microscope that were larger than ten cells. Images of colonies were taken using GelCount™ (Oxford Optronix, Abingdon, UK).

### 2.4. MTT Cytotoxicity Assay

Cell viability was determined using an MTT cytotoxicity assay according to manufacturer’s protocol. The stock solution (5 mg/mL) of thiazolyl blue tetrazolium bromide (MTT; Sigma, M2128, St. Louis, MO, USA) was prepared in DPBS (pH 7.4) and filtered to remove crystals and sterilize the solution. In a 96-well plate, 2000 cells per well were seeded and left overnight to attach. The cells were then treated with carvedilol in a serial dilution. The compounds were incubated for 24, 48 or 72 h. The MTT solution was added to each well in an amount equal to 10% of the cell culture volume, and the plate was incubated at 37 °C for 4 h. Isopropanol with 0.1N HCl (100 µL) was added to each well to replace the media before using a spectrophotometer at 570 nm wavelength to read the optic density of the formazan salt produced (reference 630 nm).

### 2.5. Dual Luciferase Reporter Assay

The BEAS-2B cells were seeded at 1 × 10^5^ cells/well in a 96-well plate in K-SFM complete media. At 80% to 90% confluency, cells were transfected with pRL-TK-Luc Renilla luciferase and pGL4.43[luc2P/XRE/Hygro] (purchased from Promega, Madison, WI, USA) plasmids at a 1:10 ratio using FuGENE HD transfection reagent (Roche Applied Science, Indianapolis, IN, USA). For ELK-1 reporter assay, pELK1-Luc (Signosis, Santa Clara, CA, USA) plasmids were transfected together with pRL-TK-Luc Renilla plasmid. After 24 h post-transfection, media was replaced by supplement-free K-SFM, and cells were co-treated with carvedilol and 10 µM B(a)P for 24 h. Cells were lysed with passive lysis buffer (Promega, Madison, WI, USA), and the firefly luciferase activity was measured using a dual luciferase reporter assay kit (Promega, Madison, WI, USA). Measurements were performed using a single mode luminometer (GloMax® 20/20 Luminometer, Promega, Madison, WI, USA). The ratio of firefly luciferase to Renilla luciferase was normalized to the negative control for AhR/XRE or ELK-1 promoter activity.

### 2.6. Real-Time RT-PCR Analysis 

BEAS-2B cells were seeded at 1 to 2 × 10^5^ cells/well in a 6-well plate with K-SFM complete media. Once the cells reached 80% to 90% confluence, the complete media were replaced by supplement-free K-SFM for 18 to 24 h, and then cells were co-treated with 5 uM carvedilol with or without 10 µM B(a)P for 6 h. Total RNA was isolated using a RNeasy mini kit (Qiagen, Hilden, Germany). cDNA was obtained using a high capacity cDNA reverse transcriptase kit (Thermo Fisher Scientific, Waltham, MA, USA). The cDNA was amplified with Power SYBR™ green PCR master mix (Applied Biosystems) and primers for human *CYP1A1* and beta-actin. PCR was performed on a CFX96 real-time thermal cycler detection system (Bio-Rad, Hercules, CA, USA). The data were analyzed based on the 2^−ΔΔCt^ method using beta-actin as the normalization gene. Primer sequences: (CYP1A1, Forward 5′-TGGTCTCCCTTCTCTACACTCTTGT-3′, Reversed 5′- ATTTTCCCTATTACATTAAATCAATGGTTCT-3′and BETA-ACTIN Forward 5′-ACCAACTGGGACGATATGGAGAAGA-3′; Reverse 5-CGCACGATTTCCCTCTCAGC-3′).

### 2.7. Animal Studies

All animal studies were approved by the Western University of Health Sciences Institutional Animal Care and Use Committee (IACUC). Mice used in the present study had access to water and food ad libitum and were housed on a 12-h light/dark cycle in a temperature-controlled facility with 35% humidity.

### 2.8. Acute Short-Term B(a)P Exposure Mouse Study

Male CD1/IGS mice (Charles River, Strain code 022), 6–8 weeks old, were used in the acute B(a)P-induced lung toxicity study. The mice were randomly assigned into the following groups with five animals in each and were treated with the following agents: (1) vehicle control; (2) a single dose of B(a)P (125 mg/kg in corn oil) given by oral gavage on the 7th day to induce lung toxicity; (3) the positive control drug curcumin (100 mg/kg, dissolved in corn oil) was given by oral gavage from day 1 to day 7, and B(a)P was given on 7th day orally after 2 h of gavaging of curcumin; (4) Carvedilol (20 mg/kg) was given by oral gavage from day 1 to day 7, and B(a)P was given on 7th day orally after 2 h of the carvedilol gavaging. Twenty-four hours after the B(a)P treatment, animals were euthanized. Blood samples were collected via cardiac puncture. The lungs were perfused with saline before being harvested.

### 2.9. Lactate Dehydrogenase (LDH) Activity Assay

LDH assay was performed using lactate dehydrogenase activity assay kit (MAK066, Sigma-Aldrich) to confirm tissue damage induced by B(a)P, following the manufacturer’s protocol using 25 μL of mouse plasma. The concentration of LDH was expressed as miliunits/mL according to reported methods [27].

### 2.10. Lipid Hydroperoxide (LPO) Assay 

Lipid peroxidation was determined using a thiobarbituric acid reactive substances (TBARS) assay (10009055, Cayman Chemical, Ann Arbor, MI, USA) according to the protocol provided by the manufacturer. As a product of lipid peroxidation, the concentration of thiobarbituric acid-reactive species was expressed as malondialdehyde (MDA) in μM according to reported methods [28].

### 2.11. Long-Term Lung Tumorigenesis Study in Mice

Female A/J mice (Jackson Lab), 8 weeks old at the beginning of the experiment, were randomly divided into the following five groups: (1) negative control group without B(a)P exposure (n = 5); (2) B(a)P-only group (n = 17); (3) positive control group exposed to B(a)P and treated with gefitinib (n = 16); (4) exposed to B(a)P and treated with 3.2 mg/kg carvedilol (n = 16); and (5) exposed to B(a)P and treated with 20 mg/kg carvedilol (n = 16). The drug treatment in groups 3, 4 and 5 started three weeks before the B(a)P exposure. The dose of gefitinib was 400 mg/kg, based on previous work [29], in which gefitinib was given by oral gavage in corn oil once a week until the end of experiment. 

Two doses were examined for carvedilol. The higher dose (20 mg/kg) was determined based on a previous study on examining oral carvedilol in chemical carcinogen 7,12-dimethylbenz(α)anthracene (DMBA)-induced skin hyperplasia in mice [21]. A more clinically relevant lower dose (3.2 mg/kg) was included as using FDA-recommended formulation for converting equivalent drug dosage between species, 3.2 mg/kg/day is equivalent to 16 mg per day in humans [30]. As the carvedilol-modified diet was not accepted by mice in a pilot mouse study, carvedilol was added into the drinking water. Carvedilol was dissolved in DMSO as stock solutions (111 and 18 mg/mL) and then diluted to the working concentrations into the drinking water. The acidified mouse drinking water (Innovive, M-WB-300A) (2.5–3.0 pH) was used to prevent the spread of bacterial disease [31], in which carvedilol showed increased solubility (>0.2 mg/mL) than in water with pH of 6.5~7.8 (5.8–51.9 μg/mL) [32]. The drug level calculation was based on previous reports [33,34]. In brief, the amount of carvedilol added into the mouse drinking water is based on the following two parameters: (1) average daily water consumption by mouse and (2) the targeted dose of carvedilol to be given to each mouse. As an example, a mouse of 25 g body weight needs 0.08 mg per day to achieve the targeted lower dose of 3.2 mg/kg/day or total amount of 0.5 mg drug per day for higher dose of 20 mg/kg/day. It was estimated that a 25 g mouse drinks ~4.5 mL liquid daily. The drug was diluted in the drinking water to a final concentration of 0.018 or 0.111 mg/mL to give targeted doses of 3.2 and 20 mg/kg, respectively. The drinking water was changed once every week. The average water consumption per cage was measured weekly, and drug concentrations in the mouse water bottles were adjusted accordingly. Animals under treatment were provided ad libitum access to drug-containing water as the only source of drinking fluid. 

For groups 2~5, to induce lung tumor, the mice were given a single dose i.p. injection of B(a)P at 100 mg/kg in 0.2 mL tricaprylin, based on published protocols [35]. At the end of the tumorigenesis studies (20 weeks after B(a)P dosing), mice were anesthetized using isoflurane. The lung of each mouse was perfused by saline and then harvested, weighed and fixed in 10% formalin for histological analysis. The surface lung tumors were counted and measured for individual tumor diameter under a dissecting microscope. Tumor volume was calculated based on the formula V = 4πr^3^/3. The total tumor volume in each mouse was calculated from the sum of all surface tumors. 

### 2.12. Histology and Immunohistochemistry (IHC) Analysis 

The formalin-fixed lung tissues were embedded in paraffin, which were de-paraffinized using xylene and ethanol. For histological analysis, the de-paraffinized sections were stained with hematoxylin and eosin (H&E). For IHC analysis, antigen retrieval was conducted by boiling the sections in an antigen retrieval buffer (Abcam Cat No. ab93678, Cambridge, UK) for 20 min. Expression of specific proteins in lung tissue was determined by IHC using a Vectastain Elite ABC universal plus kit (PK-8200) according to manufacturer’s protocol. In brief, sections were incubated with the bloxall endogenous enzyme blocking solution for 10 min to quench endogenous peroxidase activity, followed by rinsing three times (5 min each) in TBST buffer (0.05% Tween-20). Blocking solution was applied for 20 min before sections were incubated with primary antibodies overnight at 4 °C in a humid chamber: anti-COX-2 (Cell Signaling, 1:200, Danvers, MA, USA), anti-Ki-67 (Cell Signaling, 1:400, Danvers, MA, USA), anti-caspase-3 (Cell Signaling, 1:1000, Danvers, MA, USA) and anti-GADD45A (Abcam 1:500, Cambridge, UK). Ki-67 was scored by counting the positively stained cells in 7 to 12 fields at various locations per group on mouse lung tissue (n = 5 mice). Lung sections were examined and imaged using a Leica DM750 LED biological microscope.

### 2.13. Statistical Analysis

Data are expressed as individual data points with a line representing mean ± SD or ± SEM. Data presentation and error quantification are described in the figure legends. All data were graphed using GraphPad Prism version 7.03 (La Jolla, CA, USA), which was also used to analyze the data via ANOVA. For all statistical analyses, the groups were considered statistically significant when *p* < 0.05, and denotation of the statistical difference is described in the figure legends.

## 3. Results

### 3.1. Effects of Carvedilol on BPDE-Induced Malignant Transformation of BEAS-2B Cells

As an initial step to examine our hypothesis regarding the preventive activity of carvedilol against B(a)P-induced lung carcinogenesis, we used an anchorage-independent growth assay in the soft agar to evaluate the malignant transformation of BEAS-2B cells. BEAS-2B is a non-tumorous human bronchial epithelial cell line which is sensitive to transformation by B(a)P or BPDE [26,36]. BPDE at 0.2 µM significantly induced colony formation (Figure 1A,B), and, therefore, was used in the soft agar assay to examine the cancer preventive activity of carvedilol. As shown in Figure 1A, carvedilol inhibited BPDE-induced cell transformation in a dose-dependent manner. At the concentrations of 1.0 and 5.0 μM, carvedilol significantly attenuated the number of colonies in soft agar compared to the BPDE-only control. An MTT assay on the BEAS-2B cells confirmed that concentrations at 10 µM or lower did not significantly affect cell viability at up to 72 h of incubation (IC_50_ value was estimated as 25 µM for the 72-h time point) (Figure 1C). 

### 3.2. Effects of Carvedilol on B(a)P-Induced ERK-1, AhR and NF-κB Signaling in BEAS-2B Cells

Previous studies demonstrated that carvedilol attenuated epidermal growth factor (EGF)-induced ERK nuclear translocation in epidermal cells JB6 P+ and activation of ERK-regulated ETS transcription factor (ELK-1) in HEK-293 cells [37]. ELK-1 is a transcription factor that regulates gene expression via the serum response element (SRE) in the promoter regions and its activation in response to mitogen-activated protein kinases (MAPKs) [38]. As ELK-1 has been associated with lung carcinogenesis [39,40], a dual luciferase assay was used in the BEAS-2B cells to evaluate the ELK-1-regulated SRE activity. As can be seen from Figure 2A, B(a)P at 10 μM strongly induced ELK-1 activation, while the treatment with carvedilol (5 μM) reversed the activity of the ELK-1/SRE to the same level of the negative controls (Figure 2A). 

It has been reported that carvedilol inhibited UV radiation- or TNF-α-induced NF-κB activation in HEK293 cells [22]. The dual luciferase assay was used in the BEAS-2B cells to evaluate NF-κB promoter activity. B(a)P at 10 μM significantly activated the NF-κB, while carvedilol at 5 μM attenuated the activation, although not down to the same level of the negative controls (Figure 2B). Carvedilol alone (5 μM) did not affect the NF-κB activity. 

B(a)P-stimulated MAPK activation has been shown to be essential for the induction of AhR [6]. AhR is a B(a)P-activated transcription factor that regulates genes, including cytochrome P4501A1 (CYP1A1), for an increased metabolic activation of B(a)P [41]. To examine the impact of carvedilol on B(a)P-induced AhR signaling, we used the XRE luciferase assay. The data show that B(a)P at 10 μM significantly upregulated the AhR/XRE activity, while carvedilol at 5 μM, but not 10 µM, significantly attenuated its activation (Figure 2C). We next examined the mRNA expression of CYP1A1 using real-time RT-PCR. BEAS-2B cells were exposed to 10 μM B(a)P in the presence or absence of carvedilol (5.0 µM), and the cells were collected 6 h later. B(a)P strongly induced CYP1A1 mRNA (Figure 2D), while carvedilol significantly inhibited the upregulation of CYP1A1. 

### 3.3. Effects of Carvedilol on Short Term B(a)P-Induced Lung Toxicity in Mice

To determine the effects of carvedilol on B(a)P-induced lung toxicity in vivo, a single dose B(a)P-induced short-term lung toxicity study in mice was used. The experimental design is outlined in Figure 3A. In the male CD1/IGS mice, carvedilol was administered (20 mg/kg body weight, oral gavage) once a day for 7 days. The mice were treated with a single dose of B(a)P via oral gavage (125 mg/kg body weight) on day 7, 2 h after the drug treatment. Mice were sacrificed 24 h after B(a)P treatment. As expected, B(a)P increased the plasma levels of lactate dehydrogenase (LDH) and malondialdehyde (MDA) as a marker for lipid peroxidation (LPO). Treatment with carvedilol at 20 mg/kg significantly attenuated B(a)P-mediated increases in LDH and MDA levels to a similar degree as curcumin (100 mg/kg, positive control) (Figure 3B,C). 

The histological analysis (H&E) of the lung section indicates that B(a)P caused dramatic inflammatory cell infiltration, interstitial and alveolar edema, vascular congestion and alveolar collapse, while the lungs from mice treated with carvedilol exhibit a reduction in these histopathologic changes (Figure 3D). The immunohistochemical (IHC) analysis indicates that B(a)P strongly induced the expression of COX-2, caspase-3 and GADD45α, while treatment with carvedilol and curcumin attenuated these lung toxicity markers (Figure 3D). These data indicate that carvedilol effectively attenuates B(a)P-induced short-term lung toxicity, DNA damage and inflammation, which have been indicated as precursor events leading to lung carcinogenesis [42]. 

### 3.4. Effects of Carvedilol on B(a)P-Induced Lung Carcinogenesis in Mice

To further examine the chemopreventive efficacy of carvedilol, we used an established lung carcinogenesis model in A/J mice induced by a single dose B(a)P. The experimental protocol is shown in Figure 4A. Gefitinib at 400 mg/kg, given by weekly oral gavage, was used as a positive control treatment. Two drug doses of carvedilol were examined: 20 mg/kg (H) and 3.2 mg/kg (L). The mice began drug treatments 3 weeks before B(a)P treatment. To induce lung tumors, mice were given a single intraperitoneal (i.p.) injection of B(a)P at 100 mg/kg body weight. The experiment was terminated at 20 weeks after B(a)P treatment. During treatment, there was no body weight loss for all the dosing groups, except that the injection of B(a)P slightly reduced the weight, which returned to normal in a week (Figure 4B). Animal death occurred in the B(a)P-only group (one mouse died in the 9th week, and one died in the 20th week) and the gefitinib group (1 death occurred in the first week of drug treatment, possibly due to gavage error). No animal death occurred in the carvedilol treatment groups. Tumor multiplicity was determined by counting surface tumors in the fixed lung under a dissecting microscope. As expected, B(a)P significantly induced lung tumor formation (Figure 4C). Carvedilol at both doses significantly attenuated the surface tumor counts to a similar degree as gefitinib (Figure 4C). The volume data for the lung surface tumors showed similar changes (Figure 4D). The tumor multiplicity was decreased by 41.5, 43.6 and 50.7%, and the tumor load was decreased by 74.8, 66.1 and 54.3%, by gefitinib and high and low dose carvedilol, respectively.

The representative microscopic images of H&E-stained lung cross-sections showed distinct tumors in B(a)P-exposed lungs (Figure 5A). The quantification of tumor numbers in a cross-sectional area indicates the same changes as the lung surface tumor counts (Figure 5B). In the lungs of mice treated with gefitinib or carvedilol (both doses), the Ki-67 positive cells were significantly reduced compared to the mice exposed to B(a)P alone (Figure 6A,B). 

## 4. Discussion

The objective of the current study was to evaluate the effect and explore the potential mechanism of carvedilol on lung cancer development using in vitro and in vivo models induced by B(a)P, which is a PAH ubiquitously present in cigarette smoke and polluted air. First, we examined carvedilol’s effects on BPDE-induced transformation and several B(a)P-stimulated oncogenic signaling in the non-tumorous human bronchial epithelial cell line, BEAS-2B. Carvedilol showed a dose-dependent inhibitory effect on BPDE-induced transformation at 1 and 5 μM, which are both non-toxic concentrations (Figure 1). Carvedilol, at 5 μM, showed significant inhibitory effects on B(a)P-activated ELK-1, NF-κB and AhR signaling pathways, indicated using dual luciferase assays (Figure 2). The in vitro effects of carvedilol were reproduced on short- and long-term mouse models involving B(a)P. In mice, carvedilol not only attenuated B(a)P-induced lung toxicity but also reduced long-term B(a)P-induced lung cancer development. For the first time, the current study demonstrates that a β-blocker can directly inhibit the malignant transformation of lung epithelial cells and prevent lung toxicity, inflammation and carcinogenesis. 

The in vitro studies based on the BEAS-2B cell culture demonstrate a protective activity of carvedilol against the malignant transformation of human bronchial epithelial cells. BEAS-2B, an immortalized cell line with an SV-40/adenovirus-12 hybrid virus construct, has been shown as a suitable in vitro model to study lung carcinogenesis [36]. The transformation of BEAS-2B can be induced by either B(a)P, BPDE [26] or heavy metals [43,44]. However, the mechanism of BEAS-2B transformation remains obscure. It is known that B(a)P stimulates the activation of mitogen-activated protein kinases (MAPK) signaling, which has been linked with lung transformation and carcinogenesis [6,45,46]. Previous studies showed that carvedilol blocked ERK nuclear translocation and activation [37]. Consistently, in the current study, carvedilol attenuated the promoter activity of ELK-1, which is inducible by B(a)P (Figure 2). Carvedilol also inhibited the AhR pathway which was determined using an AhR/XRE luciferase assay and RT-PCR assay for the AhR-regulated *CYP1A1* mRNA in BEAS-2B cells (Figure 2). B(a)P-stimulated MAPK activation is essential for the induction of AhR [6]. The inhibition of B(a)-induced ERK phosphorylation decreased *CYP1A1* induction and AhR nuclear translocation [45,47]. Therefore, it is possible that carvedilol inhibits the AhR pathway via blocking the activation of ERK and ELK-1. As AhR is an established molecular target for lung carcinogenesis [41,48,49,50], future work should decipher how carvedilol interacts with the AhR signaling pathway. 

Previous studies indicate carvedilol’s anti-inflammatory activity. For example, based on a dual luciferase assay in HEK-293 cells, carvedilol at a concentration of 5 µM blocked UV- or TNF-α-induced NF-κB promoter activation [22]. In the murine epidermal cell culture JB6 P+, 5 µM carvedilol reduced the UV-induced production of Prostaglandin E_2_ (PGE_2_), one of the major lipid mediators of inflammation. The topical application of carvedilol (10 µM) attenuated UV radiation and induced the upregulation of the pro-inflammatory genes *COX-2*, *IL-1β* and *IL-6*. NF-κB-regulated inflammation is involved in B(a)P-mediated lung toxicity and carcinogenesis [7]. Consistent with carvedilol’s anti-inflammatory effects, in the current study, carvedilol treatment attenuated B(a)P-induced NF-κB luciferase activity in BEAS-2B cells (Figure 2). In addition, carvedilol effectively decreased the expression of COX-2 and other inflammatory markers in mice lungs induced by short-term B(a)P (Figure 3).

To reproduce the chemopreventive efficacy of carvedilol in vivo, both short- and long-term mouse models induced with B(a)P were applied. In a single dose B(a)P-induced short-term study in mice, carvedilol demonstrated protective activity against B(a)P-induced acute lung tissue oxidative damage and inflammatory biomarkers such as LDH, MDA, COX-2, caspase-3 and GADD45α (Figure 3). This study confirms that carvedilol’s cancer preventive activity may involve its anti-inflammatory and antioxidant effects. In this study, carvedilol (20 mg/kg) showed the same degree of protective effects as curcumin (100 mg/kg), which is known to show protective effects against lung inflammation [51]. The long-term lung carcinogenesis data further prove the chemopreventive activity of carvedilol (Figure 4, Figure 5 and Figure 6). However, carvedilol at two doses 3.2 and 20 mg/kg did not show dose-dependent effects, while both doses showed comparable effects with the positive control drug gefitinib. Gefitinib was used as a positive control in this long-term protocol due to its known chemopreventive activity on the same mouse model of A/J mice [29]. Gefitinib is a tyrosine kinase inhibitor targeting the epidermal growth factor receptor(EGFR)/MAPK, used as a molecularly targeted therapy for EGFR mutation positive lung cancers [52]; it may share the same mechanism as carvedilol on the MARK pathway. Further studies should determine the optimal treatment doses and duration for carvedilol. 

An alternative mechanism for carvedilol involves its known antioxidant activity [53]. In a previous study using the non-cancerous breast epithelial cell line MCF-10A cells, carvedilol at concentrations of 1, 5 or 10 µM significantly inhibited the B(a)P- (1 µM) induced formation of reactive oxygen species (ROS) formation, oxidative DNA damage and PI3K/AKT signaling [24]. Similarly, carvedilol treatment at the concentration of 10 µM inhibited the ROS formation induced by UV radiation or hydrogen peroxide in the murine epidermal cell line JB6 P+ [54]. Previous studies have also indicated that carvedilol can modulate the genes of DNA damage signaling pathways such as *RPA1*, *GADD45a* and *OGG1* [54]. As it is known that B(a)P and BPDE are genotoxic chemicals that can induce oxidative DNA damage, the antioxidant carvedilol likely prevents cancer initiation through its activity against oxidative stress and DNA damage. Among all the β-blockers, carvedilol is one of a few with antioxidant activity. It remains to be determined whether other β-blockers with or without antioxidant activity can have the same lung cancer preventive activity. 

Carvedilol is a multi-functional agent. Its mechanism of action in the lung may involve multiple targets. Its activity on inflammation and carcinogenesis further supports the hypothesis that there is cross-talk between these two events. Its activity on MAPK, AhR and NF-κB suggests that these signaling pathways are inter-connected and may be under the control of a common regulator, which remains to be identified. 

## 5. Conclusions

In conclusion, carvedilol prevents B(a)P-induced lung toxicity, inflammation and carcinogenesis via targeting multiple oncogenic signaling pathways, including MAPK, AhR and NF-κB, and, therefore, represents a promising drug candidate for use in lung cancer chemoprevention. As carvedilol is an FDA-approved drug that has been used for decades with safety records, the data from the present study suggest that a retrospective clinical trial should be designed to examine if individuals with an increased risk of lung cancer, such as smokers, who were prescribed carvedilol developed fewer cases of lung cancer than smokers who did not take carvedilol. 

## 6. Patents

US Utility Patent Application (Serial No. 17/676,684) was filed on 21 Feburary 2022 for the application of carvedilol in cancer chemoprevention

## Figures and Tables

**Figure 1 cancers-15-00583-f001:**
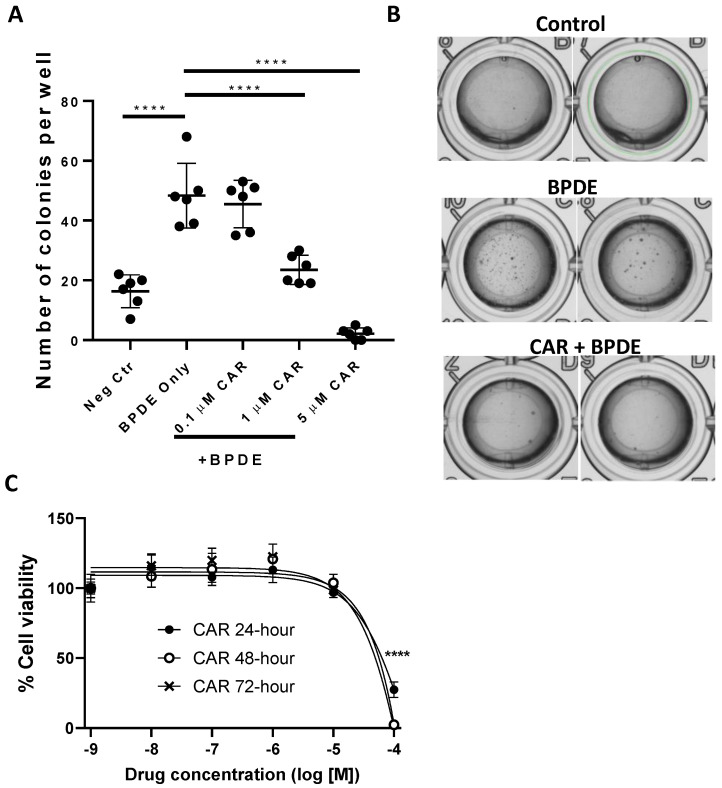
Effects of carvedilol on BPDE-induced lung epithelial transformation. (**A**) The active metabolite of B(a)P, benzo[a]pyrene diol epoxide (BPDE), was used to induce transformation of BEAS-2B cells. The cells were pretreated with carvedilol for 2 h and then treated with 0.2 μM of BPDE for 1 h, and the cells were cultured in the presence of carvedilol for 7 days. The cells were then seeded in soft agar in 96-well plate (2000 cells/well) with carvedilol in the top layer of agar. Cell colonies were counted under the microscope after 10 days of incubation. (**B**) Representative images were taken using GelCount™ of the wells containing colonies growing on agar after 10 days incubation. (**C**) Cell viability of BEAS-2B after being treated with carvedilol at various doses for 24, 48 or 72 h based on MTT assay. Data plotted are mean ± SD; n = 6. An ordinary one-way ANOVA followed by Dunnett’s multiple comparison test was used to assess statistical differences. ****: *p* < 0.0001.

**Figure 2 cancers-15-00583-f002:**
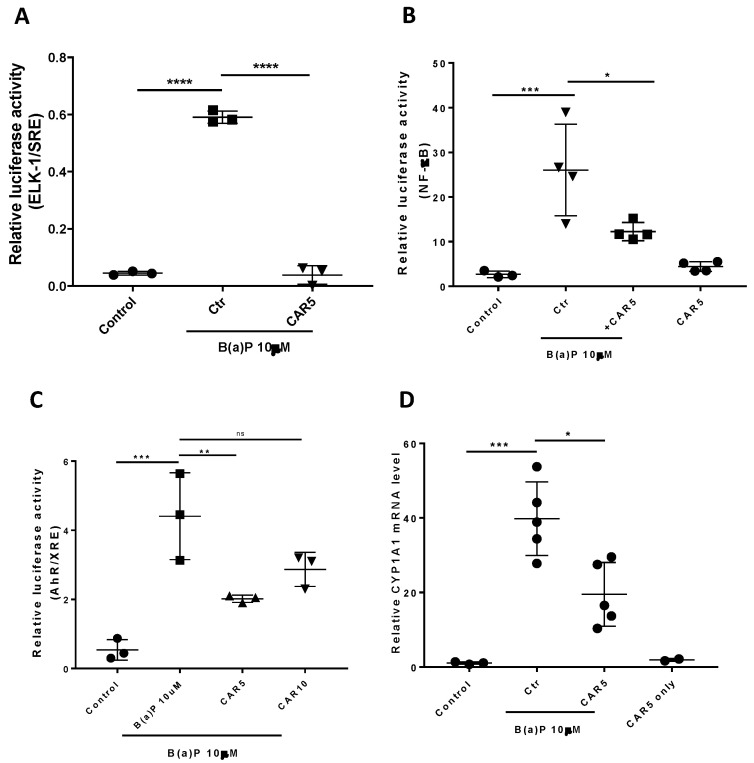
Effects of carvedilol on B(a)P-induced activation of ELK/SRE, NF-κB and AhR/XRE and expression of CYP1A1 mRNA. (**A**) Effects of carvedilol on B(a)-induced activation of ELK-1/SRE promoter. The BEAS-2B cells expressing exogenously transfected ELK-1/SRE luciferase reporter and the Renilla control reporter (10:1) were treated with vehicle, 10 µM B(a)P, with or without carvedilol for 24 h before dual luciferase assay. (**B**) Effect of carvedilol on B(a)P-induced NF-κB promoter activity. The BEAS-2B cells expressing exogenously transfected NF-κB luciferase reporter and Renilla control reporter (10:1) were treated with vehicle, 10 µM B(a)P, with or without carvedilol for 24 h before dual luciferase assay. (**C**) Effects of carvedilol on B(a)P-induced AhR/XRE promoter activity. The BEAS-2B cells expressing exogenously transfected AhR/XRE luciferase reporter and Renilla control reporter (10:1) were treated with vehicle, 10 µM B(a)P, with or without carvedilol for 24 h before dual luciferase assay. (**D**) Effect of carvedilol on B(a)P-induced mRNA expression of CYP1A1 determined using qRT-PCR. The BEAS-2B cells were treated with B(a)P (10 µM), with or without carvedilol (5.0 µM), for 6 h, before RNA isolation. Data plotted are mean +/− SD. An ordinary one-way ANOVA followed by Dunnett’s multiple comparison test was used to assess statistical differences. *: *p* < 0.05; **: *p* < 0.01; ***: *p* <0.001; ****: *p* < 0.0001. ns: non-significant.

**Figure 3 cancers-15-00583-f003:**
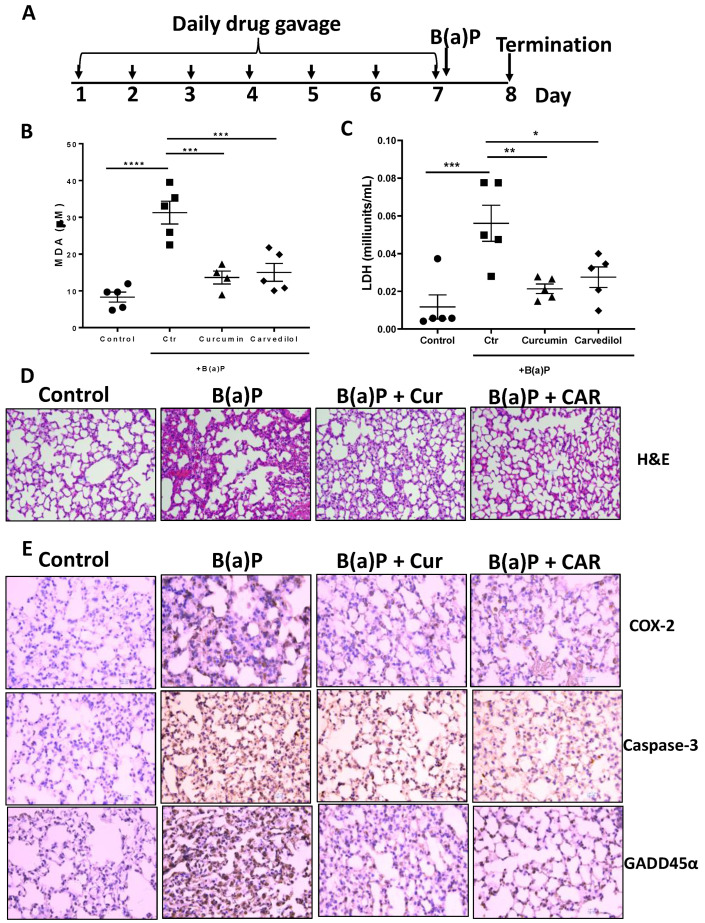
Effects of carvedilol on B(a)P-induced lung toxicity. (**A**) Experimental design. The CD-1/IGS mice were pretreated with carvedilol (20 mg/kg/day) or curcumin (100 mg/kg/day) by oral gavage for 7 days before B(a)P exposure. The mice were euthanized 24 h after B(a)P exposure. (**B**) Effects of carvedilol and curcumin on B(a)P-induced plasma levels of malondialdehyde (MDA), a marker to assess lipid peroxidation (LPO). (**C**) Effects of carvedilol and curcumin on B(a)P-induced plasma levels of lactate dehydrogenase (LDH), a marker for tissue damage. (**D**) Effects of carvedilol on B(a)P-induced histopathologic changes. H&E staining of mouse lung sections of the lung (×20). Scale bar: 100 μm. (**E**) IHC analysis for expression of COX-2, caspase-3 and GADD45α (×20). Scale bars: 100 μm. Data plotted are mean + SD. An ordinary one-way ANOVA followed by Dunnett’s multiple comparison test was used to assess statistical differences. *: *p* < 0.05, **: *p* < 0.01, ***: *p* < 0.001, ****: *p* < 0.0001.

**Figure 4 cancers-15-00583-f004:**
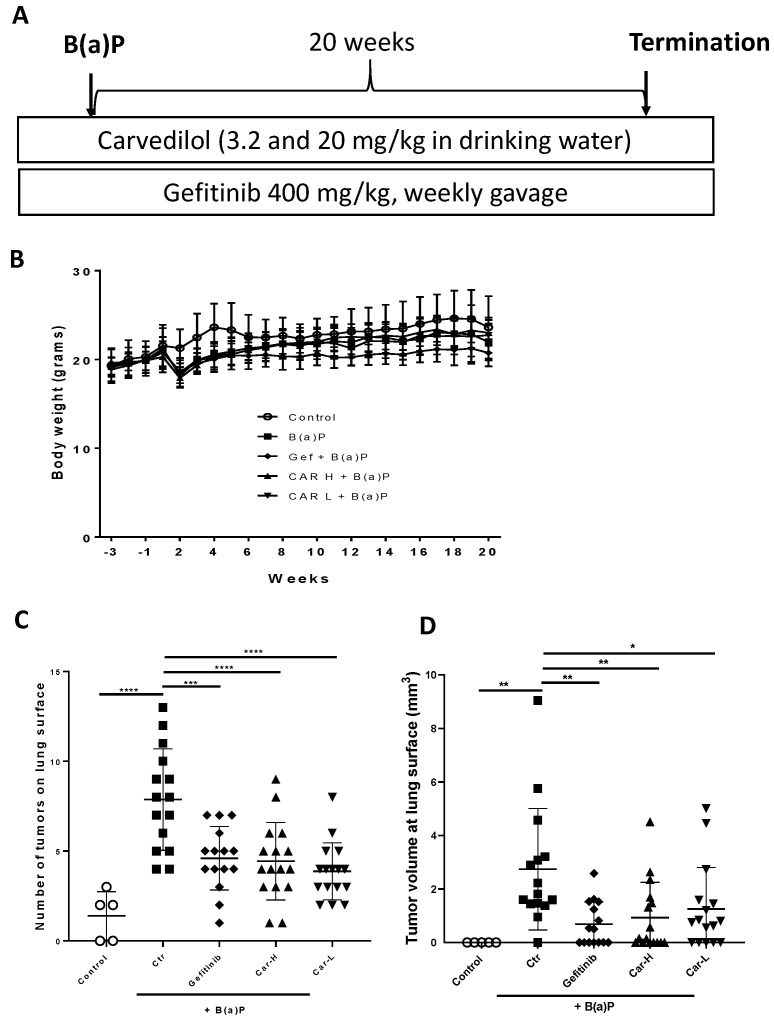
Effects of carvedilol on B(a)P-induced lung carcinogenesis. (**A**) Experimental design. The A/J mice were divided into five groups: (1) negative control group without B(a)P exposure (n = 5); (2) B(a)P-only group (n = 17); (3) B(a)P and treated with gefitinib (n = 16); (4) B(a)P and treated with 3.2 mg/kg carvedilol (n = 16); and (5) B(a)P and treated with 20 mg/kg carvedilol (n = 16). The drug treatment started three weeks before the B(a)P exposure. The dose of gefitinib was 400 mg/kg, given by oral gavage once a week. Carvedilol was given in the drinking water. (**B**) Body weight changes over the course of the studies. (**C**) Count of tumor numbers on the lung surface. (**D**) The tumor volume at the surface of the lung. Data are expressed as mean ± SD. An ordinary one-way ANOVA followed by Dunnett’s multiple comparison test was used to assess statistical differences. *: *p* < 0.05, **: *p* < 0.01, ***: *p* < 0.001, ****: *p* < 0.0001.

**Figure 5 cancers-15-00583-f005:**
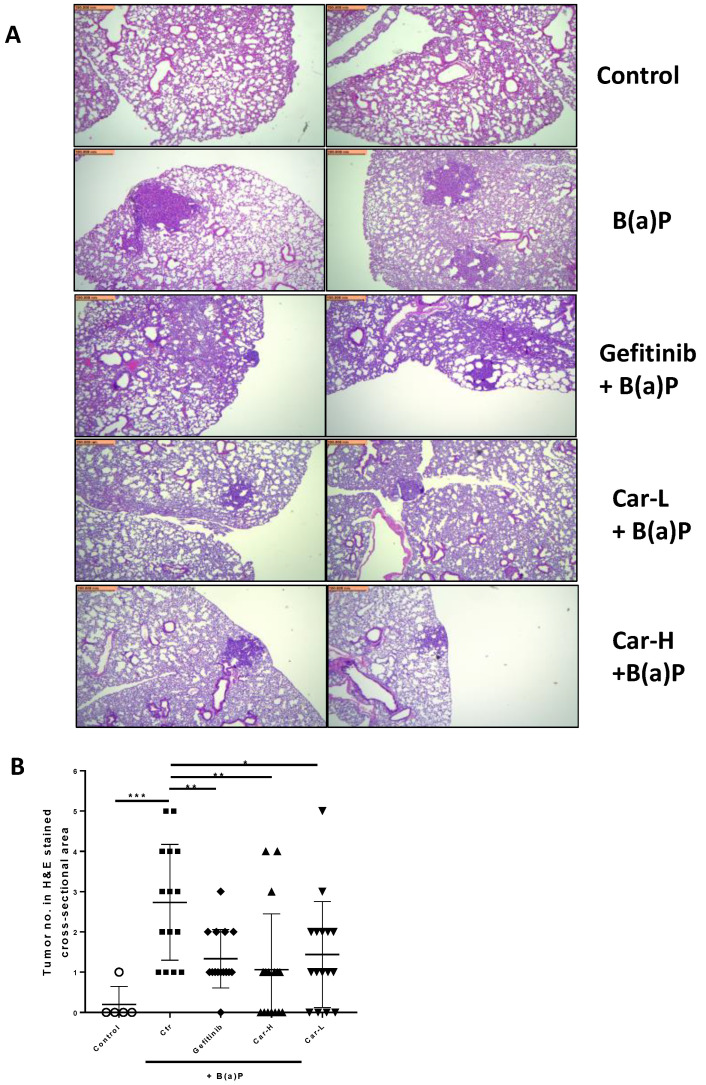
Effects of carvedilol on B(a)P-induced lung carcinogenesis. (**A**) Representative H&E-stained cross-sections of lungs from each treatment group (×4). Scale bars: 500 μm. (**B**) Number of tumors in H&E-stained cross-sectional areas of each block. Data are expressed as mean ± SD. An ordinary one-way ANOVA followed by Dunnett’s multiple comparison test was used to assess statistical differences. *: *p* < 0.05, **: *p* < 0.01, ***: *p* < 0.001.

**Figure 6 cancers-15-00583-f006:**
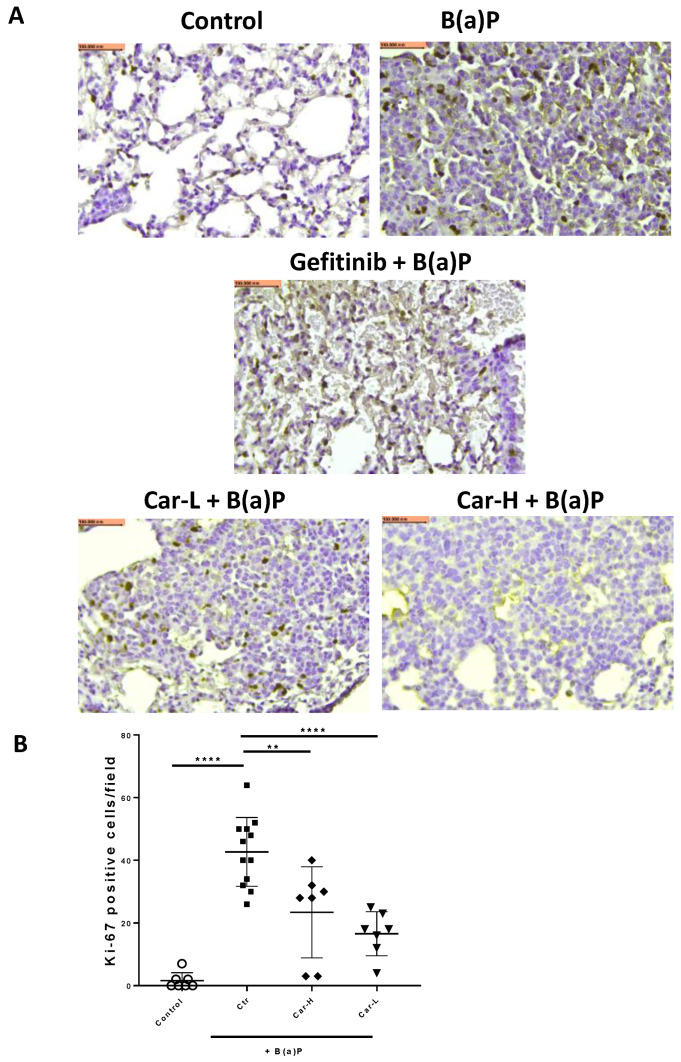
Effects of carvedilol on B(a)P-induced Ki-67 expression in lung tumors. (**A**) Representative cross-section images of lung after immunohistochemical staining with Ki-67 antibody (×20). Scale bars: 100 μm. (**B**) Number of Ki-67 positive cells in lung tumors per microscopic field. Data are expressed as mean ± SD. An ordinary one-way ANOVA followed by Dunnett’s multiple comparison test was used to assess statistical differences. **: *p* < 0.01, ****: *p* < 0.0001.

## Data Availability

The data presented in this study are available on request from the corresponding author.

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
