# Peer review of "The β-Blocker Carvedilol Prevents Benzo(a)pyrene-Induced Lung Toxicity, Inflammation and Carcinogenesis"

_cancers, 2023, doi:10.3390/cancers15030583_

Round 1

Reviewer 1 Report

  1. How did the authors select the dose and time for Carvedilol? 
  2. In figure 1A, clearly mention 2 “ctr’. This is confusing.
  3. The formatting of the manuscript is poor, specially for the figures.
  4. Why did the authors used curcumin? Please discuss briefly in the introduction.
  5. Figure 3D: should be divided into 2 separate parts-one for H & E and another for the IHC.
  6. Figure 5: Were the lungs inflated before fixation? The images are of poor quality. Please repeat the experiment (if possible).
  7. The authors must study some signaling pathways in mice. Western blot and qPCR are recommended.
  8. The manuscript should be checked for some grammatical errors.

Author Response

Reviewer 1:

How did the authors select the dose and time for Carvedilol? 

Since this is the first study investigating the effects of carvedilol on lung carcinogenesis, the doses and treatment length were mainly derived from previous studies on other types of cancer. For the in vitro studies, the doses (0.1, 1 and 5 μM) were selected based on previous studies in murine epidermal cell line JB6 P+ (PMID: 25367979). For the in vivo studies, the higher dose (20 mg/kg) was determined based on a previous study of examining oral carvedilol in another chemical carcinogen 7,12-dimethylbenz(α)anthracene (DMBA) induced skin hyperplasia in mice (PMID: 25367979). A more clinically relevant lower dose (3.2 mg/kg) was included in the long-term study, as using FDA-recommended formulation for converting equivalent drug dosage between species, 3.2 mg/kg/day is equivalent to 16 mg per day in human (PMID: 27057123). This statement has been added to the Methods (2.11, page 5). Carvedilol treatment began one or three weeks before benzo(a)pyrene exposure for examining its preventive effects. Future studies will test whether treatments after carcinogen exposure have any effect.

In figure 1A, clearly mention 2 “ctr’. This is confusing.

Change was made according to the comment. The “Ctr” was labeled as “BPDE only” in Figure 1A.

The formatting of the manuscript is poor, specially for the figures.

We tried our best to improve the formatting of the figures.

Why did the authors used curcumin? Please discuss briefly in the introduction.

Since this is the first study examining carvedilol’s effects on lung toxicity, curcumin was included in the lung toxicity mouse study as a positive control drug. Previous study has shown curcumin’s protective effects on the similar mouse model (PMID: 21859361).

Figure 3D: should be divided into 2 separate parts-one for H & E and another for the IHC.

Changes were made in Figure 3 according to reviewer’s comment.

Figure 5: Were the lungs inflated before fixation? The images are of poor quality. Please repeat the experiment (if possible).

We acknowledge that the quality of staining should be improved in our future studies.

The authors must study some signaling pathways in mice. Western blot and qPCR are recommended.

We acknowledge that signaling pathways should be studied in vivo and these studies are currently at the planning stage.

The manuscript should be checked for some grammatical errors.

We have checked the manuscript carefully for errors before the resubmission.

Reviewer 2 Report

The is a very well-conducted study that looked into the effect and explore the potential mechanism of carvedilol on lung cancer development using in vitro and in vivo models. This study also reproduces the chemo preventive efficacy of carvedilol in vivo, both short-term and 460 long-term mouse models induced with B(a)P were applied. The findings could have potential important implications for preventing lung cancer timely and for the discovery of preventative drugs.

The only concern I have or more explanation is needed is regarding using Gefitinib as a positive control. Why using Gefitinib rather than other EGFR TKI as the positive control? More explanations are needed to help readers better understand the study.

Author Response

Since this is the first study examining carvedilol’s effects on lung carcinogenesis in A/J mice, we read literature to search for a positive control drug that showed positive preventive effects against cancer development in the same model. After the literature search, we decided to include gefitinib in the lung carcinogenesis study as a positive control drug, since previous study conducted by Dr. Ming You’s group has shown its protective effects on the same mouse model (PMID: 29069801).